# Duchenne Dilated Cardiomyopathy: Cardiac Management from Prevention to Advanced Cardiovascular Therapies

**DOI:** 10.3390/jcm9103186

**Published:** 2020-10-01

**Authors:** Rachele Adorisio, Erica Mencarelli, Nicoletta Cantarutti, Camilla Calvieri, Liliana Amato, Marianna Cicenia, Massimo Silvetti, Adele D’Amico, Maria Grandinetti, Fabrizio Drago, Antonio Amodeo

**Affiliations:** 1Heart Failure Clinic-Heart Failure, Heart Transplant, Mechanical Circulatory Support Unit, Department of Pediatric Cardiology and Cardiac Surgery, Heart and Lung Transplant, Bambino Gesù Children’s Hospital, IRCCS, 00165 Rome, Italy; erica.mencarelli@opbg.net (E.M.); liliana.amato@opbg.net (L.A.); marigra525@gmail.com (M.G.); antonio.amodeo@opbg.net (A.A.); 2Pediatric Cardiology and Cardiac Arrhythmias/Syncope Unit, Department of Pediatric Cardiology and Cardiac Surgery, Bambino Gesù Children’s Hospital, IRCCS, 00165 Rome, Italy; nicoletta.cantarutti@opbg.net (N.C.); camilla.calvieri@opbg.net (C.C.); marianna.cicenia@opbg.net (M.C.); mstefano.silvetti@opbg.net (M.S.); fabrizio.drago@opbg.net (F.D.); 3Neuromuscolar Disease, Genetic and Rare Disease Research Area, Bambino Gesù Children’s Hospital, IRCCS, 00165 Rome, Italy; adele2.damico@opbg.net; 4Department of Cardiovascular and Thoracic Sciences, Fondazione Policlinico Universitario A, Gemelli IRCCS, 20097 Rome, Italy

**Keywords:** duchenne muscular distrophy, dilated cardiomyopathy, heart failure

## Abstract

Duchenne muscular dystrophy (DMD) cardiomyopathy (DCM) is characterized by a hypokinetic, dilated phenotype progressively increasing with age. Regular cardiac care is crucial in DMD care. Early recognition and prophylactic use of angiotensin converting enzyme inhibitors (ACEi) are the main stay therapeutic strategy to delay incidence of DMD-DCM. Pharmacological treatment to improve symptoms and left ventricle (LV) systolic function, have been widely implemented in the past years. Because of lack of DMD specific drugs, actual indications for established DCM include current treatment for heart failure (HF). This review focuses on current HF strategies to identify, characterize, and treat DMD-DCM.

## 1. Introduction

Dilated cardiomyopathy (DCM), arrhythmias, and congestive heart failure (HF) represent the most important life-limiting condition in Duchenne muscular dystrophy (DMD) [1,2,3]. 

Routinely cardiovascular evaluation including echocardiography is recommended in the current 2018 DMD Care consideration sponsored by Centers of disease control and prevention [4]. Moreover, HF treatments have evolved tremendously since 1980s and the armamentarium of adult HF specialists has been enriched with new drugs and the use of device (i.e., cardiac resynchronization therapy, intracardiac defibrillator, and ventricular assist device) before cardiac transplant. DMD patients are not usually candidate for heart transplantation because of the progressive skeletal myopathy, limited functional capacity [5], and shortage of donor availability.

In this review, we present the cardiologist perspective on current data regarding clinical management of DMD patients.

## 2. Pathophysiology of DMD-DCM

DMD is an X-linked recessive disorder occurring in one in 3500 male births. It is caused by mutations in the dystrophin gene that result in marked reduction or absence of the sarcolemmal protein dystrophin. 

DMD belongs to the group of dystrophinopathies, characterized by different pathogenic conditions and variable degrees of skeletal and cardiac muscle impairment. Typically, DMD is the most severe form while Becker muscular dystrophy (BMD) is the more benign form along with the X-linked DCM (XL-DCM) [6,7] and the cardiomyopathy of DMD/BMD carriers [8]. 

Several patho-mechanisms are involved in the cellular damage initially caused by the lack of dystrophin, in both skeletal and cardiac muscles. Normally dystrophin provides structural support for the myocyte and sarcolemmal membrane by its linking of actin at the C amino-terminus with the dystrophin-associated protein complex and sarcolemma at the carboxyl-terminus and the extracellular matrix of muscle [9,10]. Dystrophin is also present in T-tubular membranes of cardiac myocytes. Thus, it is involved in the maintenance of membrane stability and in the transduction of mechanical force from the sarcomeres to the extracellular matrix. The absence of the dystrophin leads to an extreme vulnerability of the cellular membranes; cellular stress could be directly mediated by the lack of dystrophin, or indirectly via intracellular Ca^2+^ overload or oxidative stress. The activation of these damaging cellular pathways and Ca^2+^ signaling pathways lead to dystrophic DCM [11]. As muscle disease progresses, skeletal and cardiac myocytes necrotize and mechanisms of repair are not adequate, with consequent progressive replacement by fibrofatty tissue [12].

DMD-DCM is characterized by thinner left ventricle (LV) wall and progressive LV dilatation, reflecting the ongoing myocyte loss [1,5]. In particular the repetitive mechanical stress leads to apoptosis and fibrotic substitution and scarring that proceeds from the epicardium to the endocardium, starting generally at the region behind the posterior and mitral valve apparatus. This scarring spreads downward progressively toward the apex and around the heart, ultimately leading to DCM [13,14].

### Clinical Course

Typically, this devasting disease is characterized by progressive skeletal muscle waste, with loss of ambulatory capacity and decline of respiratory and cardiac functions. The onset of muscle weakness is at 7–12 years old and the patient become wheel chair bound at 13 years [15]. The standard use of non-invasive home ventilators and the advance in respiratory care has changed the prognosis and prolonged survival [16]. DCM can occur at any age but often presents around 14–15 years and is very common in patients over 18 years of age [17]. It remains asymptomatic for many years in spite of the progression of cardiac dysfunction, because energy expenditure and oxygen consumption are severely diminished by muscle weakness. The degree of skeletal muscle weakness does not correlate with the severity of cardiomyopathy in patients, individual evaluation is important in order to tailor therapy and clinical assessment. Connuck et al. [5] demonstrated that the mortality rate for DMD patients with cardiomyopathy is significantly worse than that of BMD patients (who often undergo transplant) and similarly aged myocarditis and idiopathic DCM patients. The echocardiographic analysis on clinical course showed that the progression of the cardiomyopathy is slower in DMD when compared to Becker or other forms of cardiomyopathy. 

## 3. Cardiovascular Management

Current 2018 DMD Care Considerations assessed that regular cardiac assessment is essential for DMD care [4]. From the time of DMD diagnosis, every effort should be focused to detect the early onset and the progression of the DCM. Early recognition is also crucial for therapy, conditioning the life expectancy. In non-ambulatory, asymptomatic patient serial evaluation is necessary to assess the progression of the disease. Clinical evaluation remains challenging because most of these patients have often low blood pressure values, cool extremities because of reduced skeletal muscular mass. Therefore, these clinical features require multiparametric evaluation in order to differentiate whether the cardiac process is ongoing. 

## 4. Cardiovascular Biomarkers

Several biomarkers are currently used in the diagnosis and monitoring of cardiac disease. 

Electrocardiogram and cardiac imaging are routinely used to detect the onset of DCM and its progression [5]. These non-invasive tests provide useful information about left and right ventricular function, both systolic and diastolic. 

In addition, serum biomarkers provided to be very useful to characterize HF and are currently used to assess the functional status in adult and pediatric patients. In particular, serum levels of cardiac troponin I and T are known to be associated to the extension of myocardial damage, but there are conflicting results about their diagnostic and prognostic implications in the DMD DCM [18,19,20]. Recently, Voleti et al. [21] demonstrated that troponin I levels were significantly elevated in subjects with mild late gadolinium enhancement (LGE) compared to those without LGE. Interestingly, there was a lack of positive association between troponin levels and moderate-to-severe LGE probably because of a decreased enzyme leak at later stages of the disease, when most of myocardium has already been substituted by fibrofatty tissue. Hence, Troponin I could provide useful information to monitor patients in the clinical practice and further studies are required [21]. 

Elevation of left atrial pressure as result of left ventricular dysfunction and pulmonary hypertension caused by impairment of the respiratory muscles are considered to be involved in the mechanism of increased values of plasma natriuretic peptide in patients with DMD. A moderate or marked elevation in plasma alpha-ANP levels in patients with terminal DMD were found as a sign of a poor prognosis and may be a useful index for the management of the disease [22]. Villa et al. [23] reported a significant correlation between cystatin C, eGFR with cardiac dysfunction, providing for the first time a novel marker to evidence cardio-renal syndrome in patients with DMD.

## 5. Imaging

### 5.1. Transthoracic Echocardiography

Echocardiography plays the main role in identifying LV myocardial dysfunction and serial evaluation is necessary. Regional abnormalities of LV function may be revealed by early other imaging modalities such as speckle tracking echocardiography or cardiac magnetic resonance, before the overt LV dysfunction, assessed by echocardiography [24].

By baseline echocardiography, a dilated LV has been defined in terms of standard deviations, assessed with Z-scores according to the wide variability in the age and body mass index in DMD patients. In particular, LV dysfunction is defined by a LVEF < 55% and a fractional shortening (FS) < 28% [25,26]. Correlation of 2D and 3D echo techniques for LV diastolic (LVEDV) and systolic volumes (LVESV) was significantly positive, although 3D LVEDV and LVESV were lower when compared to 2D results; meanwhile LV ejection fraction estimation resulted similar by the two methods [27]. However, fractional shortening (FS) has been considered the best surrogate of LV systolic function, with respect to LVEF, for its high reproducibility [28]. FS showed a greater intraclass coefficient (ICC) than LV EF, not depending by age and magnitude of measures [28]. Regarding diastolic function, increased mitral A-wave velocities and lower E/A ratio, lower DTI lateral peak E-wave velocities were observed in DMD patients, compared to age-matched controls [28]. Another early marker of cardiac LV dysfunction in DMD patients is the myocardial performance index (MPI) obtained using both pulse-wave Doppler (PWD) and Doppler tissue imaging (DTI). On the basis of intraclass coefficient correlation (ICC), MPI obtained with DTI was more reproducible. Speckle tracking echocardiography is a technique able to evaluate subclinical LV dysfunction before development of overt LVEF reduction and has been increasingly used in DMD patients. Myocardial strain, obtained by 2D speckle tracking echocardiography is abnormal in nearly 50% of DMD patients, showing lower global longitudinal strain (GLS) values compared to healthy children, despite a normal LVEF [29,30]. Moreover, a decrease of 0.34% per year of GLS in DMD patients according to age has been recently reported [31]. In this prospective multicenter cross-sectional study, a difference in longitudinal, radial, and circumferential strain, respectively of 3.6%, 9%, and 3.8% between DMD children and matched control subjects was observed with significantly lower values in the inferolateral and anterolateral mid-basal segments [31]. Several retrospective studies previously analyzed circumferential and longitudinal strain in DMD patients, using 2D speckle tracking, with a larger magnitude of difference for these indices [25,28]. However, speckle tracking analysis is often limited in DMD because echocardiographic image quality is poor in these patients and declines by 2.5% for each 1-year increase in age [32]. Poor echocardiographic window is due to chest deformities, lung hyperinflation, and limited mobility. A suboptimal echocardiographic quality, defined as more than 30% of segments inadequately visualized, was found in 50% of 13-years-old DMD patients and 78% of 15-years-old patients [32]. Indeed, LV ejection fraction, obtained by echocardiography, has been demonstrated to correlate poorly with cardiac magnetic resonance (CMR) [26], while two-dimensional fractional shortening and 5/6 area length LVEF correlated strongly with CMR LVEF [25]. Echocardiographic reproducibility of FS and 5/6 area length LVEF has been demonstrated in DMD patients, although seemed to underestimate LV function compared to CMR [25]. 

Right ventricular (RV) function is often preserved in DMD patients, also in presence of LV dysfunction, probably because of the reduced afterload related to respiratory improvements [33]. However, right ventricle is well studied by CMR because of its high spatial resolution and reproducible data on RV myocardial deformation. In detail, Mehmood et al. [33] reported RV normal values in patients with severe LV dysfunction, and only in few cases advanced RV dysfunction. 

### 5.2. Cardiac Magnetic Resonance

Cardiac magnetic resonance is assuming an increasingly important role in DMD DCM, for the ability to identify myocardial fibrosis. Both 2D and 3D LV echocardiographic ejection fractions have a low correlation with CMR LVEF [25], and segmental analysis underestimated wall motion abnormalities detected by CMR. Indeed, CMR, not being limited by body habitus, can provide a complete and more accurate three-dimensional analysis of global and segmental LV function if compared to echocardiography with a better reproducibility. Different clinical settings have recently proved the utility of CMR in DMD patients, such as to stratify severity of myocardial involvement, or to assess the efficacy of anti-remodeling therapy in multicenter trials [34,35] to screen asymptomatic DMD female carriers [36,37], to evaluate perioperative cardiac risk. CMR allows a non-invasive myocardial tissue characterization by LGE and T1 mapping techniques, using non-ionizing radiations. The presence of a transmural LGE pattern, often located at the infero-lateral wall, is an independent predictor of adverse cardiac events in DMD patients, also in those with a preserved LVEF [38]. Furthermore, LGE pattern and distribution, ranging from subepicardial to transmural involvement, stratifies the degree of LV dysfunction severity. The presence of subepicardial LGE in the inferolateral free LV wall is a common finding in CMR of DMD patients [39] (Figure 1). In nearly 45% of DMD female carriers, a similar LGE distribution is observed [36,37,40] and it is also associated with a higher clinical class severity and myocardial enzyme release [41]. CMR is also useful to follow the development of LGE over time in DMD patients and carriers able to predict early LVEF decline, considering the higher LVEF reduction described in DMD patients with LGE, independently from age and steroid therapy [42]. However, LGE can detect focal macroscopic fibrosis, while T1 mapping technique pre- and post-contrast is able to quantify diffuse myocardial fibrosis and extracellular volume expansion (ECV). In DMD patients, T1 mapping has been demonstrated to identify early myocardial fibrosis in absence of LGE [43]. Nevertheless, significant differences in T1 mapping values were observed according to the type of T1 mapping sequence used. Previously, Soslow et al. [44] reported increased ECV values in DMD patients if compared with controls, even in cases of preserved LVEF and in the absence of LGE. Olivieri et al. [43] demonstrated the ability of native T1 mapping, by SASHA and MOLLI sequences, to stratify the presence of fibrosis also in LGE-absence. Thus, T1 mapping in DMD patients is a surrogate marker of early subclinical involvement detectable before LGE development without the need of contrast. However, T1 mapping has several limitations depending on the type of sequence used, heart rate, different vendors, and inability to discriminate diffuse myocardial fibrosis from inflammation or fat infiltration [45,46]. Finally, myocardial strain analysis may also be obtained by CMR, using feature-tracking technique. Circumferential global myocardial strain has been detected to be more impaired in DMD patients compared to controls, with more pronounced differences in anterolateral, inferolateral, and inferior segments [47]. On the contrary of 2D speckle tracking derived strain, CMR-FT is able to discriminate different values between LGE–positive and LGE-negative areas in DMD patients compared to controls, and also among segments within LGE areas [48].

Cardiac magnetic resonance in DMD DCM is important not only to detect early myocardial changes in case of subtle LV dysfunction, but also to evaluate progression of fibrosis in DMD patients on medical treatment [34,35]. However, the high costs, patient’s claustrophobia and the length of the CMR study often limit its use in many clinical centers. 

## 6. Therapeutic Strategy for DCM

Usually HF restricts the definition to the manifestation of clinical symptoms. Before clinical symptoms manifest, DCM progresses. Most of the DMD patients are asymptomatic for most of their life, so identifying precursors of the HF is crucial to manage this group. Demonstration of the ventricular dysfunction based on the assessment of ejection fraction help to guide therapy. DMD DCM really comprises a wide range of patients, from those with normal LVEF (typically considered as ≥50%) to those with reduced LVEF. Patients with an LVEF in the range of 40–49% represent a “grey area,” considered as a mid-range of DCM. In the following section, we evaluate all cardiovascular drug therapies according to LVEF [49] (Figure 2).

### 6.1. Early DCM

This group includes all cases in which LVEF is normal or ≥50%. At this stage of the disease, the aim is to delay the onset of ventricular dysfunction. Because of lack of specific therapy for DMD DCM, 2018 DMD Care Considerations [4] recommend traditional first line HF with ACE-I or angiotensin receptor blockers (ARBs). 

In 2005, Duboc [50] for the first time reported a two-phase study conducted over five years for the prophylactic use of perindopril for DMD-DCM. This study was designed to evaluate the effect of perindopril on the development and progression to LV dysfunction. In a multicenter study, 57 children aged 9.5 to 13 years with normal cardiac examination and LVEF of more than 55% at baseline as measured by radionuclide ventriculography, were randomized to perindopril 2–4 mg versus placebo. Chi-squared analysis showed a significant benefit for patients treated in order to prevent the progression of DCM, defined as reduction of LVEF below 45%. After this study, ACEi have been prescribed for prevention.

### 6.2. DCM with Mid-Range Reduction of LVEF

Few studies have been addressed for DMD patients with mid-range systolic LV dysfunction.

Current indication endorses the use of traditional HF treatment to treat the progression of the disease. In detail, for mid-range ventricular dysfunction, some studies have shown some beneficial effect to preserve ventricular function. Among ACE inhibitors, lisinopril and losartan have been used for comparative analysis in established DCM. Allen 2013 [51] compared the effects of lisinopril (an ACEi) 0.07 mg/kg (5 mg/day) with losartan (an ARB) 0.7 mg/kg (25 mg/day) in a randomized, double-blind, controlled trial of 22 DMD patients. Interestingly, if the LVEF decreased by 5 to 10% the initial dose was doubled. This trial showed no significant difference between lisinopril and losartan in preserving or improving ventricular function. 

Cardioprotective effect of adding eplerenone to an ACE inhibitor or ARB was evaluated by MRI after 12 months in 42 DMD patients. This multicenter, randomized, placebo-controlled trial, Raman et al. [35] showed that eplerenone slowed the rate of decline of magnetic resonance (MR)-assessed left ventricular circumferential strain and LVEF at 12 months, when compared to the placebo group. 

Raman et al. showed that also spironolactone added to background therapy is noninferior to eplerenone in preserving contractile function. These findings support early mineralocorticoid receptor antagonist therapy as effective and safe in a genetic disease with high cardiomyopathy risk [52].

Therefore, at the early stage of the disease, before any clinical overt DCM, the prophylactic use of perindopril for cardioprotection is entered widely in the clinical practice and endorsed by current indication although biological effects are still unclear. When the DCM is detectable even in case of mild reduction of ejection fraction (>45% LVEF), fosinopril or losartan with the combination of mineralocorticoid receptor antagonists (i.e., eplerenone) might improve ventricular function. 

In addition, beta blockers (BB) have been tested. Carvedilol was administered in 22 patients and was progressively uptitrated over 8 weeks. This therapy modestly improved cardiac MRI-derived measured ejection fraction (41% +/− 8.3% to 43% +/− 8%; *p* < 0.02), as well as the mean rate of pressure rise (dP/dt) during isovolumetric contraction (804 +/− 216 to 951 +/− 282 mmHg/s; *p* < 0.05) and the myocardial performance index (0.55 +/− 0.18 to 0.42 +/− 0.15; *p* < 0.01) [53]. 

### 6.3. Patients with Severe Ventricular Dysfunction

While in recent years, many studies have focused on the early identification of myocardial damage and the early start of cardiac therapy capable of slowing cardiac remodeling has been emphasized in DMD, the therapeutic strategy for patients with established DCM has been studied less deeply [3,16,54]. Current indication recommends all drugs used for HF treatment. 

Although in adult HF, the use of betablockers is mandatory when ventricular function declines, the same evidence in children is lacking. In recent years, some retrospective and non-randomized prospective studies have demonstrated the beneficial effect of BB therapy in patients with DMD/DCM [52,55,56,57,58], while in some others this positive effect was not observed [59,60]. Although most of the studies are retrospective including various ages, BB in adjunct to ACEi showed to improve 5-year and 7-year survival rates [58], and also improving ventricular function [56]. These conflicting results have contributed to variable and often delayed initiation of BB use in DMD. However, BBs are usually added to ACEi/ARB when a sufficient improvement in cardiac function is not achieved with the initial therapy. 

Furthermore, in DMD DCM, this therapy is often indicated for the presence of autonomic dysfunction and the consequent predisposition to arrhythmias [61].

In the current literature the drugs most frequently used are Carvedilol (0.01–0.02 mg/kg administered twice daily and slowly increased to a dose of 0.5–1 mg/kg) [53,55,56,57,59], Bisoprolol (3–4 mg per day) [58], and Metoprolol (1 to 2 mg/kg/day) [55,60].

In many studies the combination therapy with ACEi and BBs has been proved to be superior to ACEi alone in the improvement of LV function, [53,56] in the prevention of major cardiac events (death, deterioration of HF, and severe arrhythmias) [57] and in long-term survival [58].

It was noted that in patients treated with BBs the improvement of LVEF was correlated with the reduction of mean heart rate (HR) [57].

### 6.4. End Stage of DCM DMD

Our group has recently demonstrated the utility of the HR reduction (HRR) strategy obtained with BBs and Ivabradine (2.5 mg twice daily increasing until 15 mg daily every two weeks when HR was still above 70 bpm and LVEF < 40%) in the reduction of the long-term incidence of acute adverse events in DMD patients with advanced cardiac involvement [62]. Previously, ivabradine had been proven to be effective in reducing HR and in improving LVEF in a multicenter, randomized, placebo-controlled trial in children with DCM and symptoms of HF. Unfortunately in this trial DMD patients were excluded and a follow-up of only six months was considered [63].

According to European and American Guidelines for the management of HF in adults, MRAs, spironolactone, and eplerenone, are recommended in all symptomatic patients (despite treatment with an ACEI and BB) with HF and LVEF ≤ 35%, to reduce mortality and HF hospitalization [49,64].

In the near future a new MRA called vamorolone, able to mirror the anti-inflammatory effect of glucocorticoids, probably could be a valid alternative to both “old MRAs” and “simple glucocorticoids” in the DMD therapy scenario [65]. To date there are no studies about the use of MRAs in advanced phase of DCM in DMD patients. Despite this, eplerenone or spironolactone are currently used in these patients, at the discretion of the cardiologist, in addition to therapy with ACEi and BB, as long as they do not have renal insufficiency and hyperkalemia.

Sacubitril/valsartan, the first-in-class angiotensin receptor neprilysin inhibitor (ARNI) that in the past decade has changed the treatment of adult HF, has recently been approved by the Food and Drugs Administration (FDA) for the treatment of pediatric patients (aged 1 to 18 years) with symptomatic HF and systemic LV systolic dysfunction. This approval was based on the major reduction in the value of N-terminal pro-B-type natriuretic peptide (NT-proBNP) that was observed in sacubitril/valsartan arm compared to enalapril one after 12 weeks of therapy from the ongoing 52-week PANORAMA-HF trial [66]. Of note, DMD patients are included in the trial.

### 6.5. Symptomatic Drugs

Furosemide is the most common loop diuretic used to reduce systemic and pulmonary congestion and the correlated symptoms in the advanced stage of disease. For chronic use, 1 to 6 mg/kg of frusemide in partitioned doses are used. The addition of metolazone (0.1 mg/kg dose bis-in-die up to max 20 mg/day) may be useful in patients who are unresponsive to loop diuretic agents alone [67].

It is important to remark that loop diuretics are symptomatic medications and there is no evidence of their effectiveness in improving long-term prognosis [68].

Digoxin has been a pivotal drug in the treatment of HF in children and also reported as standard therapy for treatment of DCM DMD [5]. Today its use has decreased significantly in favor of more effective and safe drugs such ACEi and BBs [67,69].

In patients with severe LV dysfunction an antithrombotic therapy should be considered in the primary prevention of thromboembolic events, although not routinely recommended [70].

## 7. Advanced Cardiac Therapies

### 7.1. Heart Transplant and Mechanical Assist Device

A possible treatment for end-stage HF in these patients is the use of left ventricle assist device (LVAD) as a destination therapy (DT) [71,72]. Recently, one patient has been transplanted after 47 months of Heart Ware L-VAD assistance and after accurate respiratory and orthopedic workup. After three months the follow-up was uneventful [73]. 

LVAD has been currently used in adult and pediatric population with end-stage HF as bridge to heart transplantation or as DT in selected adult patients with medically refractory HF who are not transplant candidates [74,75,76,77]. The mechanical assist devices have established their utility in increasing cardiac output and reversing end-organ damage [75,76,77,78]. LVAD therapy significantly produced a reverse ventricular remodeling through different mechanisms: reducing ventricular size, LV mass, and at microscopic level myocyte hypertrophy and improving function [79,80,81,82,83,84,85]. 

LVADs have been recently considered as a therapeutic option as destination therapy in DMD with advanced HF [86,87,88]. The use of mechanical circulatory support in DMD has been described in case reports and small series [86,87,89,90,91,92,93]. 

Selection of patients is crucial and several aspects should be considered (i.e., kyphoscoliosis, respiratory muscle weakness, and recovery and rehabilitation after surgery). Analysis of costs [94] showed that DT-VAD in DMD exceeds cost-effectiveness thresholds but was similar to cost-effectiveness estimates of DT-VAD in adults who are not transplant candidates. 

### 7.2. Ethical Aspects

Actually, end-of-life management preferences in neuromuscular diseases, including DMD, are a challenging area. Ethical concerns remain open about which patient should be a candidate or excluded. Additionally, The Working Group acknowledged the value of a long-term patient/family/physician relationship before the urgent need for device placement [72]. A multidisciplinary approach with careful evaluation of frailty and co-morbidities is crucial to assess the proper selection of DMD patients. A shared decision process is necessary to obtain a collaborative contact with patient, parents, and caregivers, making this strategy successful [95,96].

## 8. Arrhythmias in DMD

Arrhythmias occur frequently in cardiomyopathies. They may be also isolated manifestations, mostly in myotonic dystrophies and muscle channelopathies [97]. Potentially fatal arrhythmias are terminal events, and require the implantation of incatracardiac defibrillator (ICD). ECG may show right axis deviation, deep and narrow Q waves in inferolateral leads, conduction defects, sinus tachycardia, short PR intervals, and tall R wave in the right precordial leads, right bundle branch block and flat and inverted T waves [98]. In a large multicenter French study, left bundle branch block was present in 13% of patients; 2/3 of them disclosed exonic deletions. Left bundle branch block was significantly associated with cardiac events and mortality [99]. The incidence of supraventricular (6%) and ventricular arrhythmia (VT, 2%) was low in that study, in line with previous data that reported atrial flutter in 5%, sinus pause 5% [100], VT in 7% [101]. Others described that the QRS duration tended to increase progressively with age, irrespective of LV systolic function in patients with DMD [102].

### 8.1. Electrophysiologic Characteristics

Arrhythmias are observed in a mouse model of DMD after acute β-adrenergic stimulation. In men, a case reported arrhythmic storm after abrupt withdrawal of beta-blocker therapy [103]. Arrhythmia may be linked to aberrant expression and remodeling of the cardiac gap junction protein connexin43 (Cx43). Opening of remodeled Cx43 hemichannels plays a key role in the development of arrhythmias in DMD mice. Then, these channels can be therapeutic targets to prevent fatal arrhythmias in patients with DMD [104]. 

DMD patients are prone to ventricular arrhythmias, which may be caused by abnormal calcium (Ca^2+^) homeostasis and elevated reactive oxygen species. In an animal model of DMD, a susceptibility to pacing induced ventricular arrhythmias was demonstrated. Oxidated Ca^2+^/calmodulin-dependent protein kinase II, Ox-CaMKII, promotes aberrant sarcoplasmatic reticulum Ca^2+^ release through RyR2, which leads to delayed afterdepolarizations and triggered ventricular arrhythmias. Genetic inhibition of ox-CaMKII normalized intracellular Ca^2+^ and prevented ventricular arrhythmias in this model [105].

### 8.2. CRT and Implantable Cardioverter Defibrillator

It is known that DMD patients are at risk of arrhythmias (such as atrial fibrillation, atrial flutter and ventricular tachycardia) but, in the absence of dedicated studies, the DMD Care Considerations Working Group suggests to apply the standard antiarrhythmic medications and device management recommendations. At present, also the indication for the ICD is based on the established adult HF guidelines [49] but should be individualized according to clinical status, nutritional state, and respiratory function.

Cardiac resyncronization therapy (CRT) implantation improved symptoms and heart function in two DMD patients with HF and left bundle branch block. Mortality remains higher in similar DMD patients without CRT [98]. Other authors reported limited benefits with the implantation of ICD and CRT in dystrophinopathic cardiomyopathies, with no increase of EF, no change or worsening of EDV [106]. However, some patients (a quarter) had subjective improvements in their daily activities. Causes of this poor response in DMD patients could be the normality of QRS complex and the extensive postero-lateral fibrosis [107].

## 9. DMD Target Therapy

Glucocorticoid treatment has been the standard of care for patients with DMD. Prednisone and deflazacort are the most commonly recommended steroids. The introduction of steroid therapy has changed the natural history of the disease, prolonging the autonomous ambulation period, delaying the cardiorespiratory insufficiency occurrence, and increasing children life-expectancy [4,108].

The understanding of molecular basis and knowledge of DMD has led to the advent of several experimental therapeutic approaches. The therapeutic approaches for DMD have focused on restoring dystrophin expression or mitigating the processes downstream of dystrophin deficiency.

The strategies that have been embraced for dystrophin protein restoration include (1) nonsense readthrough, (2) antisense oligonucleotides for exon skipping, and (3) gene therapy. To mitigate the dystrophic processes the approaches used are (1) inhibiting inflammation, (2) promoting muscle regeneration, (3) reducing fibrosis, and (4) facilitating mitochondrial function. 

This translational research has led to the approval of first treatments for DMD and several other agent are under clinical investigation. Ataluren (Translarna™, PTC Therap.) is the first approved drug for DMD in Europe. Ataluren is an oral molecule that binds ribosomal RNA subunits and enables ribosomal readthrough of mRNA containing a premature stop codon. It is indicated for the treatment of DMD resulting from a nonsense mutation in the dystrophin gene. 

More recently FDA conditionally approved AONs targeting exon 51 (eteplirsen) and 53 (golodirsen) for the treatment of DMD [61]. 

## 10. Female DMD Carriers

DMD is an X-linked condition and it usually affects males, with the majority of females having mutation in a single allele being asymptomatic carriers. About one-third of all DMD cases are caused by de novo mutations, with the other two-thirds due to inheritance from the mother. This means that every mother of an isolated male DMD case has a two-thirds chance of being a carrier. Most females with a pathogenic DMD gene variant present as asymptomatic carriers because of the presence of a second normally functioning allele. However, some female carriers can present symptoms from mild to more severe clinical courses, as muscle weakness, abnormal gait, fatigue, and cardiac involvement [109]. These patients are classified as “manifesting carriers.” The “skewed inactivation of the normal X-chromosome” hypothesis was widely used to explain the mosaic pattern of dystrophin expression in skeletal as well as heart muscle in female DMD carriers. According to this hypothesis, a higher percentage of skewed inactivation of the normal X-chromosome was responsible for the occurrence of cardiomyopathy in some female MD carriers. However, in the study of Brioschi et al. [110], there was no relationship between the dystrophic muscular phenotype and either the X-inactivation pattern or the dystrophin transcriptional behavior, suggesting that the major cause of disease manifestation is simply the total dystrophin protein amount. Each male child of a carrier female has a 50% chance of being clinically affected with DMD. Carrier testing can indicate if a woman is at risk of having affected sons. Although female DMD carriers are mostly free of skeletal muscle symptoms, cardiac symptoms affect about 8% of this population with DCM as a common presentation. They may develop cardiomyopathy ranging from asymptomatic forms with mild abnormalities to progressive HF, even requiring heart transplantation [36]. The onset of clinical manifestations for symptomatic female carriers is variable, ranging from early childhood to late adulthood [111]. The incidence of cardiomyopathy increases with age, even in patients with normal electrocardiograms and no skeletal muscle symptoms. Therefore, in the clinical guidelines in Europe and the United States [15], adult dystrophinopathy carriers are recommended to undergo echocardiography every 5 years. Other cardiac manifestations include conduction defects and arrhythmias, but these could be consequences of long-term DCM. Acute HF and non-sustained ventricular tachycardia have been reported as initial presentations in late adulthood, although these are not as common. Disease severity is variable and genotype–phenotype correlations are not well established in this group of patients, and cardiac involvement may be present without concomitant skeletal muscle manifestations [111,112,113]. Cardiac manifestations in female carriers may be subclinical under normal physiological conditions. They can worsen and become symptomatic during major events such as pregnancy. Approximately two-thirds of all patients with limbic-type muscular dystrophy experience muscle weakness during pregnancy and these are probably related with weight gain and diaphragm elevation. 

Besides cardiac manifestations, female DMD carriers can present with other systemic features: limb girdle weakness, gait disturbance, exercise intolerance, calf hypertrophy, and scoliosis have all been recognized as skeletal muscle manifestations in these patients. Elevated serum creatinine kinase (CK) is often found in patients with skeletal muscle presentations. Additionally, neurocognitive problems can present as learning disabilities or behavioral problems in this patient population [112,113].

## Figures and Tables

**Figure 1 jcm-09-03186-f001:**
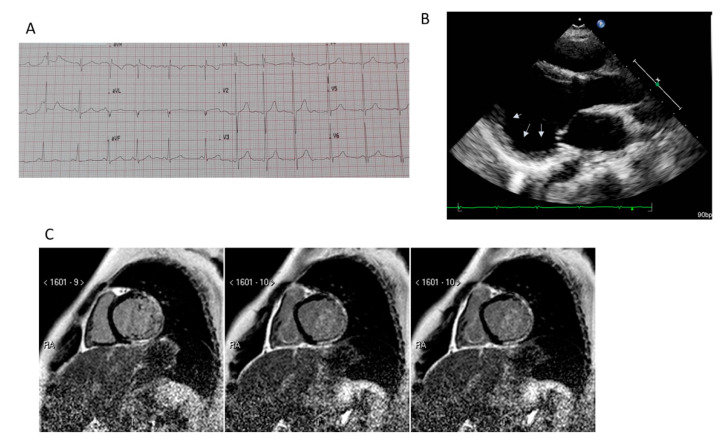
Clinical features of Duchenne muscular distrophy cardiomyopathy (DMD-DCM). Panel (**A**): typical EKG with sinus tachycardia and tall R waves. Panel (**B**): parasternal long axis view of left ventricle (LV). Arrows indicate the presence of posterior wall aneurysm. Panel (**C**): cardiac magnetic resonance: short axis view of the LV. Presence of a transmural late gadolinium enhancement pattern located at the infero-lateral wall (Courtesy of Dr. A. Secinaro).

**Figure 2 jcm-09-03186-f002:**
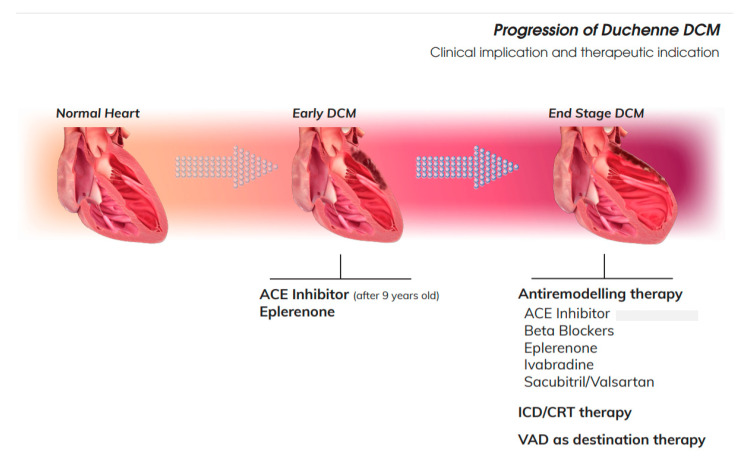
Progression of DMD-DCM. According to clinical stage of the DMD-DCM, different strategy might be considered. ACE: Angiotensin Converting Enzyme; CRT: Cardiac Resyncronization Therapy; ICD: IntraCardiac Defibrillator; VAD: Ventricular Assist Device.

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
