# Peer review of "Duchenne Dilated Cardiomyopathy: Cardiac Management from Prevention to Advanced Cardiovascular Therapies"

_jcm, 2020, doi:10.3390/jcm9103186_

Round 1

Reviewer 1 Report

Rachele Adorisio et al. present a review on duchenne dilated cardiomyopathy.

Specific comments.

  1. Before re-submission the manuscript needs extensive English editing (numerous typos, spelling errors, odd sentences and words). Therefore, I strongly suggest proof-reading by a native speaker before any re-submission. 
  2. I would suggest to re-name the "characteristics of the DMD-DCM section into "Pathyphysiology of DMD-DCM". Furthermore, it would be interesting to expand this pathophysiology section.
  3. The imaging section should be split up into sub-sections: e.g. transthorathic echocardiography, cardiac magnetic resonance imang
  4. Moreover, additional figures covering pathophysiology and imaging modalities (e.g CMR figures of various stages of the disease) would make the manuscript more reader-friendly.

Author Response

  1. Before re-submission the manuscript needs extensive English editing (numerous typos, spelling errors, odd sentences and words). Therefore, I strongly suggest proof-reading by a native speaker before any re-submission. 

Thank you for your suggestion. Two native speakers (American Language) read and corrected the paper

2. I would suggest to re-name the "characteristics of the DMD-DCM section into "Pathyphysiology of DMD-DCM". Furthermore, it would be interesting to expand this pathophysiology section.

Thank you for this suggestion. We changed the title of the section and we expanded

3. The imaging section should be split up into sub-sections: e.g. transthorathic echocardiography, cardiac magnetic resonance imang

According to this suggestion, we split up the section into subsection

4. Moreover, additional figures covering pathophysiology and imaging modalities (e.g CMR figures of various stages of the disease) would make the manuscript more reader-friendly.

According to this suggestion, we included 2 figures in the text with MRI and echo images, depicting typical features of DMD cardiomyopathy

Reviewer 2 Report

This is an important review by very experienced specialists

The reviewer has several comments and suggestions to make this paper appilcable to general cardiologists and practitioners who almost don"t encounter these patients:

  1. The paragraph "Characteristics of the DMD-DCM" should be expanded to describe the natural history of DMD-DCM developing in along the progression of the muscle disease. Some of the material from the part of therapy in the different stages of the disease may be moved here.
  2. DMD-DCM should be placed withing the spectrum of dystrophinopathies. In fact, even among patients with Duchenne,  there is quite a spectrum of severity and pace of progression.
  3. A separate paragraph may addess the difficulties of clinical assessment and obtaining adequate imaging. The discussion of CMR is academic but most of these patients cannot tolerate CMR testing. The role of sinus tachycardia in the early stage as well as specifics/examples of ECG are worthy of attention. Difficultions in drug therapy should be specifically addressed: low blood pressure, cold extremeties,very low muscle mass, tendency to develop hypokalemia  and risks of digoxin toxicity.  
  4. There is a notion that DMD DCM patients usually do not suffer from malignant ventricular arrhythmia untill advanced stages of disease (Arbustini JACC papers). Given the potentially bad effect of implanting a device on skeletal and respiratory function, and the non-ischemic  etiology, DMD DCM is a case when a conservative approach to primary prevention ICDs should  be exercises on a case by case basis.
  5. A separate paragraph whould address DMD in women
  6. A separate paragraph has to address diagnosis of DMD-DCM versus  other DMD with myopathy
  7. I miss mentioning DMD stesific therapies and their effect on the heart. Steroids need to be discussed
  8. The part  on heart transplant and LVAD is out of proportion. While there are case reports and exceptions, HTX is generally  not an option in DMD (unlike Beckers) while LVAD is rarely suitable. It is important  to put it this way  not to mislead the general clinician.

Author Response

The reviewer has several comments and suggestions to make this paper appilcable to general cardiologists and practitioners who almost don"t encounter these patients:

  1. The paragraph "Characteristics of the DMD-DCM" should be expanded to describe the natural history of DMD-DCM developing in along the progression of the muscle disease. Some of the material from the part of therapy in the different stages of the disease may be moved here.

Thank you for this comment. We changed this paragraph and included some aspects of natural history

2. DMD-DCM should be placed withing the spectrum of dystrophinopathies. In fact, even among patients with Duchenne,  there is quite a spectrum of severity and pace of progression.

Thank you for this comment. We included other dystrophinopaties and differences have been reported

3. A separate paragraph may addess the difficulties of clinical assessment and obtaining adequate imaging. The discussion of CMR is academic but most of these patients cannot tolerate CMR testing. The role of sinus tachycardia in the early stage as well as specifics/examples of ECG are worthy of attention. Difficultions in drug therapy should be specifically addressed: low blood pressure, cold extremeties,very low muscle mass, tendency to develop hypokalemia  and risks of digoxin toxicity.  

Thank you for this suggestion, we included specific comment from practical perspective, in all paragraph

4. There is a notion that DMD DCM patients usually do not suffer from malignant ventricular arrhythmia untill advanced stages of disease (Arbustini JACC papers). Given the potentially bad effect of implanting a device on skeletal and respiratory function, and the non-ischemic  etiology, DMD DCM is a case when a conservative approach to primary prevention ICDs should  be exercises on a case by case basis.

Thank you for this practical suggestion. We included specific DMD comment in clinical arena in each paragraph (page 17 and page 19), underlying the importance to tailor therapy according to clinical status, nutritional state and respiratory function

5. A separate paragraph whould address DMD in women

A separate paragraph has been added

6. A separate paragraph has to address diagnosis of DMD-DCM versus  other DMD with myopathy

We added in the text differences with other cardiomyopathy

7. I miss mentioning DMD stesific therapies and their effect on the heart. Steroids need to be discussed

We added a specific paragraph with general consideration

8. The part  on heart transplant and LVAD is out of proportion. While there are case reports and exceptions, HTX is generally  not an option in DMD (unlike Beckers) while LVAD is rarely suitable. It is important  to put it this way  not to mislead the general clinician.

We shortened the paragraph, although there Is a big debate on it

Round 2

Reviewer 2 Report

Thank you

Now this is excellent

Congratulations

It was a honor for me to contribute to this paper as a reviewer